

# Variation in *Pheidole nodus* (Hymenoptera: Formicidae) functional morphology across urban parks

Yi Luo[1], Qing-Ming Wei[2], Chris Newman[3], Xiang-Qin Huang[1], Xin-Yu Luo[1] and Zhao-Min Zhou[1,4]

[1] Key Laboratory of Southwest China Wildlife Resources Conservation (Ministry of Education), China West Normal University, Nanchong, China
[2] Nanchong Vocational and Technical College, Nanchong, China
[3] Wildlife Conservation Research Unit, Department of Biology, University of Oxford, Oxford, United Kingdom
[4] Key Laboratory of Environmental Science and Biodiversity Conservation (Sichuan Province), China West Normal University, Nanchong, China

## ABSTRACT

**Background**. Habitat fragmentation and consequent population isolation in urban areas can impose significant selection pressures on individuals and species confined to urban islands, such as parks. Despite many comparative studies on the diversity and structure of ant community living in urban areas, studies on ants' responses to these highly variable ecosystems are often based on assemblage composition and interspecific mean trait values, which ignore the potential for high intraspecific functional trait variation among individuals.

**Methods**. Here, we examined differences in functional traits among populations of the generalist ant *Pheidole nodus* fragmented between urban parks. We used pitfall trapping, which is more random and objective than sampling colonies directly, despite a trade-off against sample size. We then tested whether trait-filtering could explain phenotypic differences among urban park ant populations, and whether ant populations in different parks exhibited different phenotypic optima, leading to positional shifts in anatomical morphospace through the regional ant meta-population.

**Results**. Intraspecific morphological differentiation was evident across this urban region. Populations had different convex hull volumes, positioned differently over the morphospace.

**Conclusions**. Fragmentation and habitat degradation reduced phenotypic diversity and, ultimately, changed the morphological optima of populations in this urban landscape. Considering ants' broad taxonomic and functional diversity and their important role in ecosystems, further work over a variety of ant taxa is necessary to ascertain those varied morphological response pathways operating in response to population segregation in urban environments.

Corresponding author
Zhao-Min Zhou,
zhouzm81@gmail.com

## INTRODUCTION

The paradigm of morphological shifting remains a central topic in evolutionary biology. Traditionally, ecogeographical rules (*e.g.,* Bergmann's, Allen's and Hesse's rules) have often been used to explain spatial and/ or temporal patterns of morphological variation for endothermic (*e.g., Cui et al., 2020*; *Ryding et al., 2021*) or ectothermic (*e.g., Arnan, Cerdá & Retana, 2015*; *Guilherme et al., 2019*; *Sosiak & Barden, 2021*) species in response to gradients in environmental conditions, such as temperature. Variations in morphological traits may also result from phenotypic plasticity; that is, the capacity of a single genotype to produce a range of phenotypes under different environmental conditions (*Gibbin et al., 2017*); however, plasticity can be adaptive, non-adaptive or neutral depending on its relation to the optimal fitness in the changed environment (*Ghalambor et al., 2007*). Adaptive responses occur when the plastic response is favored by directional selection.

Environments may differ in their ecological optima (*i.e.,* a certain combination of ambient factors that is optimal for the growth, existence and reproduction of an organism), leading to directional selection and shifts in a population's position within ecological space, or environments may differ in the range of ecological variation they can support (*i.e.,* in the strength of environmental filtering they impose), constraining the volume of ecological space occupied by a species, or population (*Algar & López-Darias, 2016*). These two options are not mutually exclusive and could act in concert or opposition. Morphological shifts can occur at different taxonomic levels, but typically involve the adaptation of populations to localized environmental conditions (*Cohen, Haran & Dor, 2019*; *Cohen et al., 2021*), especially when driven by ecological isolation.

Urbanization is an increasingly important driver of global change (*Szulkin, Munshi-South & Charmantier, 2020*; *Des Roches et al., 2021*). Urban ecological factors (*e.g.,* food availability, physiological demands, habitat modification, pollution, thermal landscape) create selective pressures that can have a strong effect on remnant populations, resulting in phenotypic traits that are divergent from conspecifics living in natural areas (*i.e.,* urban-derived phenotypes, *Diamond & Martin, 2021*; *Diamond, Prileson & Martin, 2022*). For example, in urban areas, neotropical lizard (*Anolis cristatellus*) populations exhibit longer limbs relative to body size, more sub-digital scales and a greater heat tolerance, although it remains to be established if this is genetically based (*Winchell et al., 2016*; *Campbell-Staton et al., 2020*). Importantly, phenotypic changes may occur within the lifespan of the individual (*Cohen et al., 2021*).

Within an urban ecosystem, isolated 'green spaces' represent 'green islands' with the potential to preserve faunistic and floristic assemblages. City parks often differ from one another in size, structure and complexity, and thus provide highly informative crucibles for studying how individual organisms adapt their tolerances, resulting in differentiation among populations enabling them to persist within them. Evidence of differentiation is scarce, but has been found both in invertebrates (*De Carvalho et al., 2017*) and vertebrates (*Littleford-Colquhoun et al., 2017*). Furthermore, *Winchell et al. (2023)* have shown that genomic parallelism could underlie phenotypic parallelism (*i.e.,* convergence), in response to consistent urban environmental trends.

Ants are an ecologically dominant faunal group in most terrestrial ecosystems, which play key ecological roles as nutrient cyclers, predators, soil engineers and regulators of plant growth and reproduction (*Czechowski et al., 2012*). Studies on ants have informed understanding on ecological responses to disturbance and land management (*Andersen, 2019*), and how they can provide a bioindicator of environmental degradation (*Andersen, 1997*; *Andersen & Majer, 2004*).

There is an extensive literature describing how ants adapt to urban park conditions; however, this mainly focuses on diversity and the structure of ant community assemblages (*Clarke, Fisher & LeBuhn, 2008*; *Ślipiński, Zmihorski & Czechowski, 2012*; *Carpintero & Reyes-López, 2014*; *Reyes-López & Carpintero, 2014*; *Liu et al., 2019*; *Nooten et al., 2019*). In contrast, there is a scarcity of research examining whether environmental heterogeneity among isolated urban parks can cause morphological shifts within fragmented ant populations. Given, however, that morphological shifts among worker ants from various species have been attributed to various environmental drivers in populations in natural habitats, morphological shifts adapting to urban habitats seem highly plausible (*Heinze et al., 2003*; *Clémencet & Doums, 2007*; *Bernadou et al., 2016*; *Antonov, 2017*; but see *Gaudard, Robertson & Bishop, 2019*). In these previous studies, homospecific populations tended to be characterized on the basis of the mean value of their functional traits, as though these represent a static unit in space and time. Ignoring intraspecific trait variation across paratypes could underestimate a species' ability to compete with other community members (*Ashton et al., 2010*), the degree of niche and trait overlap (*Courbaud, Vieilledent & Kunstler, 2012*), resource use (*Bolnick et al., 2003*; *Eggenberger et al., 2019*) and potential responses to new conditions found in urban environments.

The prevalence of some ant generalist species in urban environments may be attributable to high functional morphology plasticity, enabling them to utilize different habitats. Examining workers of a local dominant ant species, *Pheidole nodus*, from urban parks in Nanchong city, China, here we tested for differences in how individuals from segregated populations fill morphological space (morphospace), and whether these differences arise due to the strength of environmental filtering (environmental filter-strength hypothesis). This would be implied if different phenotypic optima between park environments drive directional selection and shifts in a population's position within morphospace. Alternatively, parks may differ in the range of ecological variation they can support (optimum-shift hypothesis), constraining the volume of ecological space occupied by each ant population. Either of these mechanisms in isolation, or combined, could potentially lead to shifts in the positions of populations in morphospace between habitats.

## MATERIALS & METHODS

### Study area

This study was conducted in seven urban parks around downtown Nanchong (30°46′–31°85′N, 105°44′–106°96′E), located in Sichuan (China) (Fig. 1). This downtown area covers c. 160 square kilometers, occupied c. 1.5 million residents. The downtown core stretches north-south along the Jialing River. Nanchong has an East Asia monsoon climate,

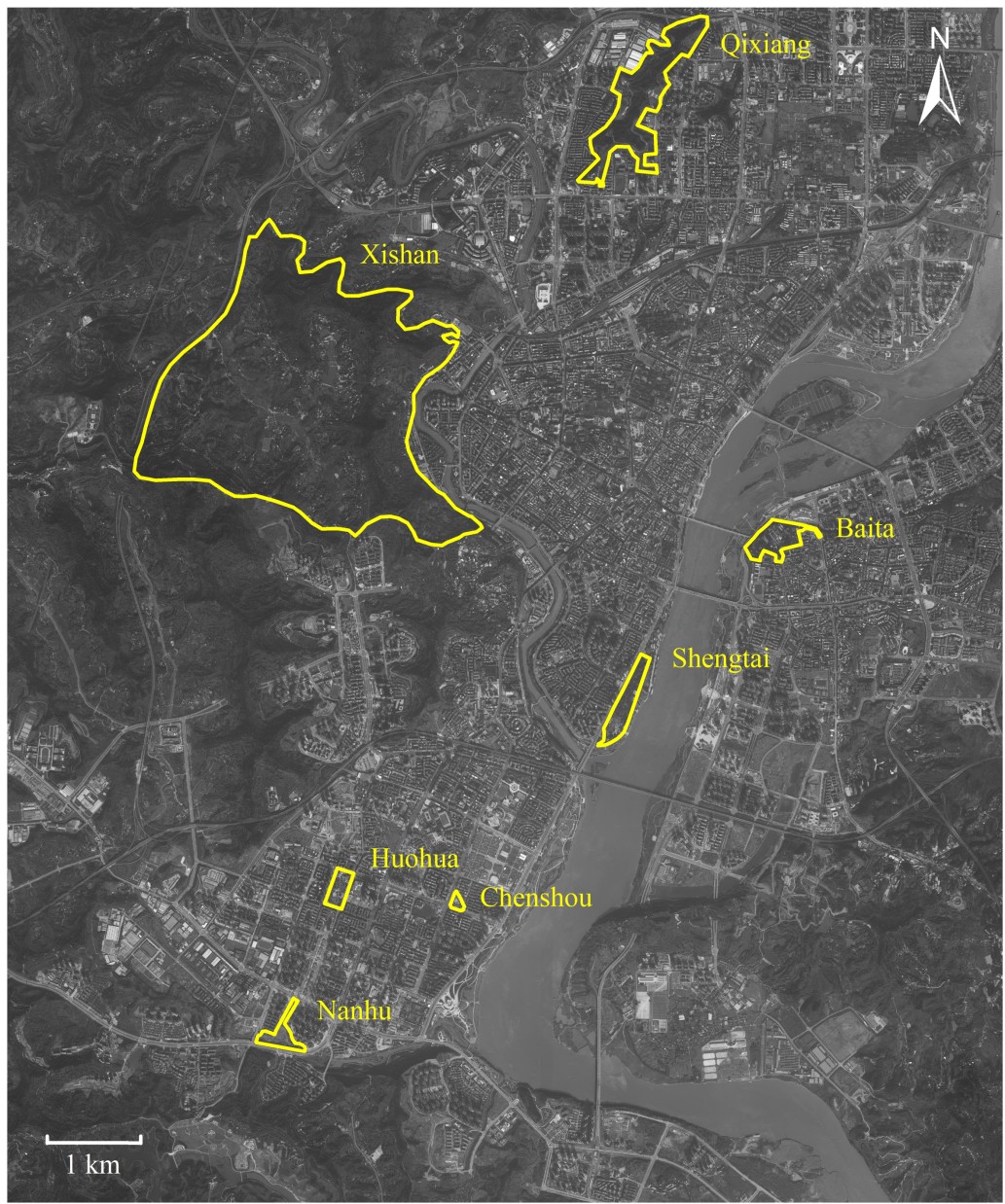

**Figure 1 Map of Nanchong City showing the seven parks studied.** Parks are highlighted with yellow borders. Map credit: Baidu Maps (https://map.baidu.com/).

with an annual rainfall of 980 to 1,150 mm, a temperature range of 15.8 to 17.8 °C, and relative humidity from 76.0% to 86.0% (*Liu et al., 2018*; *Luo et al., 2023*).

## Ant sampling

*P. nodus* belongs to the subfamily Myrmicinae; a generalized functional group that copes well with disturbance (*Hoffmann & Andersen, 2003*; *Kuate et al., 2015*), dominating ant communities in warm, shady habitats where food resources are clumped and thus defensible

(*Andersen, 1997*). We obtained foraging workers of *P. nodus* from seven urban parks (Fig. 1). We did not include workers that remain in the nest because our aim was to test the effects of environmental selection on ants actively foraging under urban habitat conditions. At six of these parks (Baita, 0.333 km$^2$; Chenshou, 0.017 km$^2$; Huohua, 0.067 km$^2$; Nanhu, 0.373 km$^2$; Qixiang, 0.700 km$^2$; and Shengtai, 0.300 km$^2$) the habitat was comprised mostly by impervious man-made surfaces, lawns and some sparse shrubs and trees planted deliberately (Fig. S1). In contrast, we also included Xishan Park; a larger park (18.8 km$^2$) designated to protect the remnant native broad-leaved evergreen forest (Fig. S1). These seven parks were distributed 1.3 to 10.8 km apart across the high-density residential downtown district.

Ants were collected using pitfall traps from July to August 2020 during the summer season; the time of year when ant activity is greatest (*Luo et al., 2023*), maximizing foraging interactions with urban habitat conditions. At each park we established 16 pitfalls with at least 10 m spacing, to include all micro-habitat types and sample over as large an area as possible. Each trap consisted of a plastic container (7.1 cm diameter × 9 cm height) containing a solution of ethylene glycol (as a killing and preservative agent) and water at a 1:2 ratio. Each trap was covered by a small plastic plate attached to three nails to limit solution evaporation and rainfall ingress. Ants from each park were collected after sampling for 7 days and stored in 95% ethanol. We did not collect foraging workers and colony workers from nests because nests could not be located reliably due to only protruding c. five cm above ground and generally being concealed under shrubbery. Sampling ants predominantly from more obvious nests could potentially lead to sampling bias. Field experiments were approved by the Research Council of China West Normal University (project number: CWNU2020D002). Foraging workers of *P. nodus* were then identified based on taxonomic keys for ant fauna (*Wu & Wang, 1995*) and for the genus *Pheidole* (*Pan, 2007*). All specimens for this study were stored and examined at China West Normal University.

## Quantifying morphological characteristics

Morphological parameters were measured by Q.-M. Wei using a Leica M205C binocular microscope system and the 'Leica Application Suite' software. Six continuous morphological traits (Table 1) with functional significance were measured for each worker ant. These raw measurement data were used in a principal component analysis with a varimax rotation to identify key axes of morphological variation. The first four principal components that accounted for 88.16% of the total variance (PC1 = 49.46%, PC2 = 15.96%, PC3 = 12.37%, PC4 = 10.37%) were retained. PCs 1-4 were loaded heavily by traits related to body size, sensory abilities, prey size and capacity to explore habitat, respectively (Table 2, Fig. S2).

## Testing the environmental filter-strength and optimum-shift hypotheses

We applied four PCs to analyze morphospace. In order to account for unequal sampling across parks, we randomly sub-sampled 10 workers from each park and then computed

**Table 1  Morphological traits, indicator of function and anatomical definition following *Gibb et al. (2015)* and *Grevé et al. (2019)*.**

| Trait | Indicator of function | Measure |
|---|---|---|
| Weber's length | Indicative of worker body size. | Distance from the anterodorsal margin of the pronotum to the posteroventral margin of the propodeum. |
| Head width | Size of gaps through which worker can pass, which is also a trait used as estimation of the worker size. | Maximum width of the head across the eyes. |
| Eye width | Indicative of the exploratory capacities of the habitat. | Measured across the maximum width of the eye. |
| Mandible length | Indicative of diet; longer mandibles could allow predation of larger prey. | Straight-line distance from the insertion to the tip of the mandible. |
| Scape length | Sensory abilities-longer scapes facilitate following of pheromone trails, finding food and perceiving vibrations, etc. | The maximum straight-line length of the scape, excluding the basal constriction. |
| Hindleg length | Leg length is the predictor for locomotory skills and habitat complexity. | Femur length and tibia length of one of the hindleg combined to leg length. |

**Table 2  Eigenvectors for the first four principal components from a principal components analysis of ant morphology after varimax rotation.**

| Variables | PC1 | PC2 | PC3 | PC3 |
|---|---|---|---|---|
| Head width | 0.889 | 0.151 | −0.029 | 0.161 |
| Eye width | 0.236 | 0.103 | 0.072 | 0.963 |
| Mandible length | 0.095 | 0.042 | 0.998 | 0.066 |
| Scape length | 0.305 | 0.938 | 0.041 | 0.104 |
| Weber's length | 0.893 | 0.165 | 0.099 | 0.140 |
| Hind leg length | 0.690 | 0.299 | 0.172 | 0.164 |

convex hulls with the 'hypervolume' package (*Blonder et al., 2014*; *Blonder, 2015*) in R ver. 4.1.1 (*R Core Team, 2021*). We repeated this sub-sampling 100 times; that is, we constructed 100 convex hulls for workers collected from each park. To assess significance, we generated a null expectation by randomly selecting 10 individuals from all the individuals collected, without replacement, and repeated this 1,000 times. Given that the null expectation exhibited a non-normal distribution, independent-sample Mann–Whitney $U$ tests were employed to assess differences in convex hull volume and morphospace centroid (*i.e.,* the distance of a population's centroid in morphospace from the overall centroid). In all analyses, significance was set at $P < 0.001$.

We also measured morphological dissimilarity (MD), which describes the percentage of total meta-population morphospace volume occupied uniquely by *P. nodus* workers from each population. To do so, we compared the MD of worker ants collected from separate parks, to include all 21 pairwise combinations, with the total MD among all parks.

To test the environmental filter-strength hypotheses, the convex hull volume for workers from each park was compared to the volume expected from a null model.

To test the optimum-shift hypothesis, the distance from each population's morphospace centroid to the centroid of all populations pooled was calculated, and compared to a null expectation.

**Table 3  The contribution of individual parks to total morphological dissimilarity, *i.e.* the percentage of total morphospace volume uniquely occupied by ants from each park.**

| Park | Unique Volume (%) | P-value |
|---|---|---|
| Baita | 20.1 | =0.048 |
| Chenshou | 0.9 | <0.001 |
| Huohua | 2.5 | <0.001 |
| Nanhu | 28.6 | <0.001 |
| Qixiang | 11.8 | <0.001 |
| Shengtai | 2.7 | <0.001 |
| Xishan | 13.1 | <0.001 |

**Table 4  Morphological turnover (% non-overlap, below diagonal) and P-values for the difference in convex hull volume (above the diagonal) among pairs of parks.**

| | Baita | Chenshou | Huohua | Nanhu | Qixiang | Shengtai | Xishan |
|---|---|---|---|---|---|---|---|
| Baita | | <0.001 | <0.001 | <0.001 | <0.001 | <0.001 | <0.001 |
| Chenshou | 99.5 | – | <0.001 | <0.001 | 0.299 | <0.001 | 0.067 |
| Huohua | 95.5 | 97.6 | – | <0.001 | 0.858 | <0.001 | 0.016 |
| Nanhu | 97.8 | 93.6 | 97.2 | – | <0.001 | <0.001 | <0.001 |
| Qixiang | 92.4 | 98.9 | 97.9 | 97.9 | – | 0.038 | <0.001 |
| Shengtai | 97.3 | 86.6 | 95.7 | 90.9 | 97.5 | – | 0.003 |
| Xishan | 91.9 | 95.4 | 93.4 | 93.3 | 96.0 | 89.6 | – |

## RESULTS

A total of 163 *P. nodus* workers were collected and measured; 19 from Baita, 15 from Chenshou, 15 from Huohua, 18 from Nanhu, 38 from Qixiang, 15 from Shengtai, and 43 from Xishan. Workers from a single park uniquely occupied 79.7% of total population morphospace dissimilarity (MD) (Table 3), which was significant based on our null model. The contribution of each individual park to MD ranged from 0.9 to 28.6%, where unique morphospace volume differed significantly from the null exception for each park except for Baita (Table 3). Among park pairs, MD ranged from 89.6 to 99.5%, and only six park population pairs (of 21; Chenshou-Qixiang, Chenchou-Xishan, Huohua-Qixiang, Huohua-Xishan, Qixiang-Shengtai and Shengtai-Xishan) occupied similar convex hull volumes (Table 4, Fig. S3).

   Morphospace (convex hull) volume clustering was significant at four of these seven parks, Chenshou (median value = 0.80), Huohua (1.14), Qixiang (2.76) and Shengtai (1.66), but was not significant for Xishan (3.86, $P = 0.975$); significantly larger convex hull volumes were evident at Baita (4.57) and Nanhu (8.50) parks. (Fig. 2A). Centroid displacement was significantly greater than the null expectation value at Baita (median population distance from pooled centroid across all populations = 0.98), Nanhu (1.25) and Qixiang (1.11) parks, but was not significant for Chenshou (0.74, $P = 0.100$) and Xishan (0.67, $P = 0.624$); significantly lower centroid displacements were evident for Huohua (0.57) and Shengtai (0.50) (Fig. 2B).

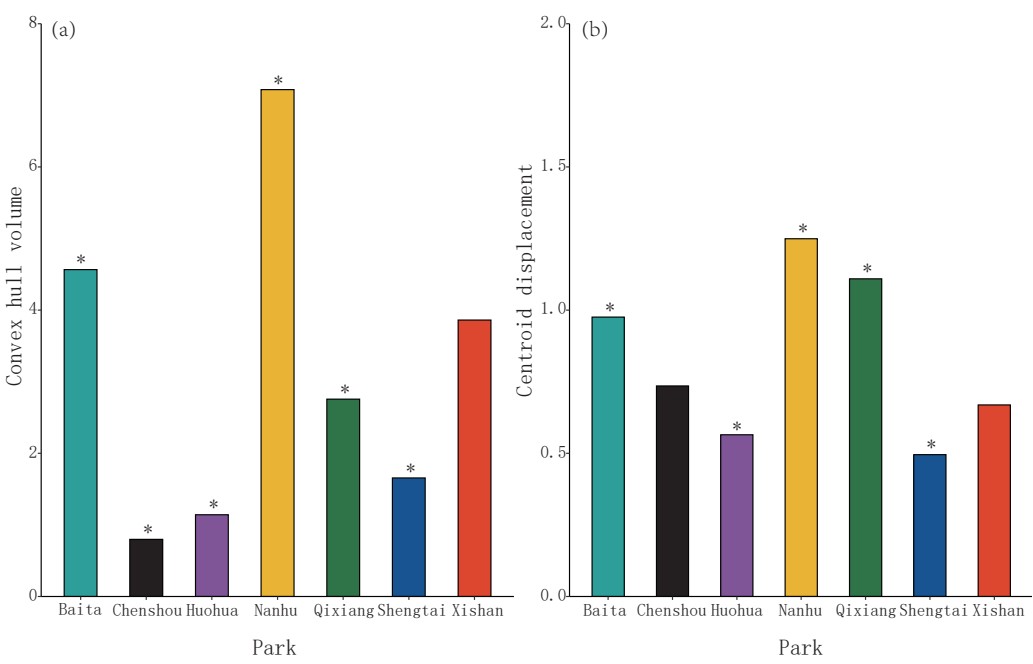

**Figure 2** **The volume and position of morphospace occupied by sub-populations of *Pheidole nodus* collected from the seven parks in Nanchong City (Fig. 1).** Convex hull volume (A) indicates the extent of trait-clustering, while centroid displacement (B) depicts the distance of each sub-population's morphospace centroid from the pooled centroid across all populations. An asterisk (*) indicates a *P*-value less than 0.001 based on the null model.

## DISCUSSION

We found that *P. nodus* populations from urban parks spaced several kilometers apart exhibited distinct morphometric traits variations. Although these seven urban populations were likely entirely segregated from each other, morphospace volumes at four of the seven urban parks were smaller than the null expectation, and centroid displacements at three of these urban parks exceeded the null expectation, suggesting trait clustering and a shift in trait optima. These results are highly consistent with the environmental filter-strength hypothesis, which proposes that the strength of environmental filtering varies across environments, leading to differential trait-clustering (phenotypic variance). Simultaneously, these results also support the optimum-shift hypothesis, which proposes that the phenotypic optimum differs between environments, leading to divergence in phenotypic morphospace position between populations in different environments (see: *Algar & López-Darias, 2016*). From this we deduce that generalist ant species, such as *P. nodus*, can exhibit both substantial intraspecific variation in phenotypic diversity and optima in response to differences in habitat conditions.

Furthermore, we also found substantial segregation for both phenotypic clustering and optima between populations. This suggests that ant inter-population dispersal *via* spillover between these urban parks was severely limited, due to habitat fragmentation. This is typical in urban landscapes, where residential, commercial and industrial development, along with

roads and paved pedestrian areas present effective dispersal barriers, especially for species with restricted dispersal distances (*Fletcher Jr, Reichert & Holmes, 2018*). Indeed, the urban landscape habitat matrix presents a generally inhospitable and ubiquitous barrier to colonization by many urban-avoiding terrestrial arthropods (*Ricketts, 2001*; *Schmidt, 2008*; *Brühl & Eltz, 2010*). Recent studies of ants conducted across Manhattan's urban habitat mosaic (*Savage et al., 2015*), as well as in the suburban riparian corridor in Ku-ring-gai Local Government Area, Sydney, Australia (*Ives et al., 2011*) and across urban parks in Taichung, China (*Liu et al., 2019*) all found similar evidence of impeded spillover between fragmented populations. As for *P. nodus* specifically, its dispersal potential is unknown but, in general, ants are poor dispersers, with genetically viscous population structures (*Sundström, Seppä & Pamilo, 2005*; *Hakala, Perttu & Helanterä, 2019*).

Land-use disturbance arising due to urbanization generally intensifies selective pressures that reduce diversity and causes biotic homogenization (*Rocha-Ortega & Castaño Meneses, 2015*; *Morelli et al., 2016*). Consequently, locally invasive non-native biotas often steadily replace native biotas. Our finding that ant populations in Chenshou, Huohua, Qixiang and Shengtai parks had more homogeneous morphological trait patterns suggests that selective pressures in these urban sites favor reduced phenotypic plasticity in corroboration of this homogenization scenario. Similar intra-specific canalization of phenotypic diversity has been noted for other taxa in urban settings, such as mosquitos (*Culex nigripalpus*) (*De Carvalho et al., 2017*), Puerto Rican Crested Anoles lizards (*Anolis cristatellus*) (*Winchell et al., 2016*; *Winchell et al., 2018*) and neotropical Wedge-billed Woodcreepers (*Glyphorynchus spirurus*) (*Thompson et al., 2022*). In contrast, urban conditions appear to drive phenotypic diversity in bumblebees (*Bombus pascuorum* and *Bombus lapidarius*), because urban landscaping and management can lead to cities having a greater morphological variety of flowering species in any one season compared to rural areas (*Eggenberger et al., 2019*).

In our study, the population at Xishan Park was exceptional, where neither morphospace volume nor centroid displacement were distinct from that expected based on random sampling across all the collected individuals. Plausibly, this may be because this park is much larger than the others we sampled and continues to support a remnant of more natural native broad-leaved evergreen forest, providing habitat diversity and a broader range of food types. That the intra-specific morphological diversity among *P. nodus* workers under Xishan Park's more natural conditions was around average, compared to the full meta-population sampled, suggests that these ants have evolved an optimal base morphology suited to their most natural habitat.

Mechanistically, it remains difficult at this stage to resolve whether these functional trait shifts are (i) an adaptive response to urban conditions, (ii) simply an example of background capacity for phenotypic plasticity, or (iii) based on genetics or epigenetics. Although *P. nodus* belongs to the generalized Myrmicinae functional group, which prefer open habitats and are often favored by disturbance (*Andersen, 2019*), the tendency for phenotypic clustering and optimum displacement in urban populations suggests that this species does not exhibit pre-adaptations (exaptation; *sensu Gould & Vrba, 1982*) to urban conditions, unlike (for example) from cockroaches, pigeons and rats. Plausibly phenotypic

characteristics differ among urban populations due to myriad novel environmental effects on ontological development. To test for a heritable genetic component would require testing the fitness of F1 and F2 offspring reared in a controlled environment (*McDonnell & Hahs, 2015*). Alternatively, if workers inherit traits from the queen that contribute to colony fitness, this would provide an adaptive explanation. Any heritable genotypic adaptation to novel environmental conditions would arise through either of two selective mechanisms: (1) the expression of phenotypic plasticity (the ability of one genotype to express varying phenotypes when exposed to different environmental conditions); and (2) evolution *via* directed selection for particular phenotypes, resulting in the modification of the population gene pool (*Clusella-Trullas, Terblanche & Chown, 2010*; *Fox et al., 2019*; *Desmond, 2021*; *Jacquier et al., 2021*; *Lambert et al., 2021*). Even in the absence of genetic data, the reduced morphospace and centroid displacement we detected are consistent with urban habitat fragmentation reducing genetic diversity within urban populations, while simultaneously increasing populations genetic differentiation from a standard neutral model. Such neutral processes are typically associated with detrimental fitness consequences, in part arising from fragmentation and population isolation of that leads to increased drift and elevated probability of inbreeding depression (*Combs et al., 2018*).

Ours is the first preliminary study to report differences in variance of ant functional morphology at species level among urban parks. Our study involved a relatively small sample size, due to the low abundance of ants in this highly disturbed region collected using a pitfall sampling method, chosen to sample morphological traits from only foraging workers experiencing urban habitat conditions, and to exclude colony workers. On the basis of this sample size, we caution that those differences in the morphospaces of each park population we detected, while indicative, should not be considered conclusive until further work builds on this foundation to more fully quantify the effects of ecological filters on intraspecific morphological variation among ants in urban micro-habitats.

## CONCLUSIONS

The evidence we found for morphological distinctiveness and segregation for both phenotypic clustering and optima between populations implies that filtering and selective mechanisms may both be operating on ant functional morphological traits. Ultimately, however, it remains to be seen whether *P. nodus* will have sufficient plasticity to continue to persist faced with the novel challenges of urban habitats, especially with concomitant climate change pressures (*Goh, 2020*), and will be robust to those Allee effects that can perturb isolated populations (*Schoereder et al., 2004*). Nevertheless, our results begin to demonstrate how novel urban environments present challenges to ant generalist populations, to which species respond through phenotypic, and possibly adaptive, change.

## ACKNOWLEDGEMENTS

Thanks to the inter-departmental members of the research team for their collaboration and assistance while gathering data for this research project.

### Funding

Yi Luo was supported by the scientific research foundation of China West Normal University (19D044). Zhao-Min Zhou was supported by the scientific research foundation of China West Normal University (17YC365) and the National Natural Science Foundation of China (31600412). The funders had no role in study design, data collection and analysis, decision to publish, or preparation of the manuscript.

### Grant Disclosures

The following grant information was disclosed by the authors:
Scientific research foundation of China West Normal University: 19D044, 17YC365.
National Natural Science Foundation of China: 31600412.

### Competing Interests

The authors declare there are no competing interests.

### Author Contributions

- Yi Luo conceived and designed the experiments, authored or reviewed drafts of the article, and approved the final draft.
- Qing-Ming Wei performed the experiments, analyzed these data, prepared figures and/or tables, and approved the final draft.
- Chris Newman conceived and designed the experiments, authored or reviewed drafts of the article, and approved the final draft.
- Xiang-Qin Huang analyzed these data, prepared figures and/or tables, and approved the final draft.
- Xin-Yu Luo performed the experiments, prepared figures and/or tables, and approved the final draft.
- Zhao-Min Zhou conceived and designed the experiments, authored or reviewed drafts of the article, and approved the final draft.

### Field Study Permissions

The following information was supplied relating to field study approvals (*i.e.*, approving body and any reference numbers):

Field experiments were approved by the Research Council of China West Normal University (project number: CWNU2020D002)

### Data Availability

The raw measurements are available in the Supplemental Files.

### Supplemental Information

Supplemental information for this article can be found online at http://dx.doi.org/10.7717/peerj.15679#supplemental-information.

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
