# Peer review of "Variation in Pheidole nodus (Hymenoptera: Formicidae) functional morphology across urban parks"

_PeerJ, doi:10.7717/peerj.15679_

## Round 0.1 · original submission · Major Revisions

Despite the study looking interesting, it can not be considered for publication at this stage. The reviewers point out some issues, especially with the sampling. Please, consider the comments of reviewers 2 and 3 seriously, when you decide to submit a new version of the manuscript.

Reviewer 1 ·

Basic reporting

The article is well written and structured and provides all information necessary to understand what the authors have done and how it fits in the academic literature.
In terms of language, I have one comments about the sentence starting in line 61: ants are the subject of this sentence, but ants do not "understand" the responses to disturbance, etc. It should be modified to make it clear that "we" as humans/ scientists have an understanding of these phenomena in ants.

Experimental design

The experiment is well designed and has not been performed in this way before. While community structure in cities has been repeatedly studied as the authors note in the manuscript, the intraspecific phenotypic diversity of ants in urban spaces has so far not been the topic of published studies. It thus contributes an important piece of knowledge about intraspecific biodiversity in urban environments.
Based on the method section and presented data, the study is well done and the procedures are described well so that replication is possible.
However, I have some concerns about the framing and hypotheses of the article which I will describe in the next section.

Validity of the findings

While the general experiment and results are well described and documented, I find myself a bit confused concerning the framing of the article and its finding in several parts of the manuscript.
Broadly, this concerns the distinction between the environmental filter-strength hypothesis and the optimum-shift hypothesis that the authors make in their article. The first focuses on an ecological process in which community assembly is based on the traits present within a population (including potential phenotypic plasticity) which are then filtered by the environment. The second is an evolutionary process in which populations shift their phenotypic traits/ morpholgy to fit a to theoretical phenotypic optimum of the specific environment they live in over macroevolutionary timescales.
The authors claim in the abstract and at the end of the introduction that they test whether their findings are better explained by either of these two hypotheses and their analysis does seem to be set up in principle to attempt this distinction. In the results section of the abstract, they imply that the pattern they found is based on environmental filtering. However, based the results of their analyses would suggest that they should rather refer this to phenotypic optima, based on how the analysis is set up. In the discussion, then, they state that their results are consistent with both hypotheses (e.g., line 176) and later, (line 215 to 217), they note that their current results are not suited to clarify whether the observed differences in phenotypic traits across subpopulations are the result of adaptation or phenotypic plasticity. Finally, in the Conclusions, the authors claim that "the occurrence of unique morphological traits implies that selective mechanisms are involved", pointing to the optimum shift hypothesis as explanation for the observed pattern.
All this leads to ambiguity for me as reader in how far these hypotheses were actually tested and if the authors ultimately favor one of them and if so, which of them.
From my personal perspective, I find it unlikely that substantial morphological evolution has happened in this ant species in urban spaces. I find the parts of the manuscript that discuss the potential reduction of phenotypic diversity of subpopulations due to bottleneck/ founder effects rather compelling (e.g., abstract line 34, 35) but these are not really followed up in any detail in the discussion.
One potential weakness of the study in regards to the two contrasting hypotheses is that the authors do not have data for the morphospace occupied by the investigated species in natural habitats. If they did, they could better argue that the morphology of some city populations is outside of the range of naturally occuring phenotypes (or within it).

In any case, I am not sure how well my thoughts about this come across, but this maybe reflects my confusion with the framing of the article. I think before the article is published, the authors should try to improve clarity in what exactly their competing hypotheses are (if they even have competing hypotheses, which they don't necessarily have to, in my opinion), what these hypotheses predict and which of them is favored by their data and analysis.
As it stands, different parts of the article seem to tell a different story regarding this.

Reviewer 2 ·

Basic reporting

This study stresses the importance of population-level studies to understand the ability of species to adapt to environmental changes such as fragmentation and urbanization. The study model and parks are promising but the sampling size seems to really be on the low side. This makes the robustness of the data an issue unless proven otherwise.
The introduction should focus on the need to obtain more population-level data on divergence among sites, which is what the authors do provide. A more thourough review of known population-level differences among sites (urban parks, urban parks/natural areas) should be provided. The underlying mechanisms that are responsible for divergences should be presentend in the discussion, as there are mutiple potential factors involved and the study cannot tell them apart.

Experimental design

L113. Were all ants from all parks collected across those 2 months ? There may be biaises in sampling caused by meteorological events or seasonality.
L114. Ants were collected using pitfall traps, i.e. you collected foragers. In this species, what is known about forager morphology in relation with workers that remain in the nest, and could this add any bias to your results ? In addition, you may have collected foragers from several colonies, or just one, and this may also heavily bias mean and diversity of shapes. How many colonies are there along your pitfall setup ? Where did you position the pitfals in the large parks ? Would sampling at nest entrance be a possibility ? How far does this species forage ? It would also be useful to compare your data on foragers with morphology of all individuals from a nest. I suggest to increase your sampling in order to make your results more robust.

Validity of the findings

L158. The number of workers collected from some parks is really low with regard to computation and comparison of morphospaces. This raises the question of whether finding one extra worker could completely shift the morphospace of one park : many datapoints are clearly isolated in Fig2, which suggests that the sampling might not properly reflect the real diversity of workers. A problem that subsampling cannot overcome. Appart from Chenshou that does seem more compact than the others on Fig2, the rest looks like it could be changed by adding just one or two outliers.
L160. What do you mean by ‘from a single park’ ?
L162 & 165 : please specify M-W statistics, df and P values in the main document and not in supplementary are this is your main result.
Fig2. It is very hard to tell datapoints apart. Maybe just showing PC1vsPC2 would be enough, with all other plots in supplementary ?

Additional comments

L59 – 65. If you wish to quote Andersen, please use quotation marks. As they are, these sentences are too similar to the original publication.
L62. Please rephrase ‘ants have a well-developed understanding of responses to disturbance’ to ‘researchers on ants have…’
L82. Add an ‘s’ to ‘habitat’
L190-191. Do you have any perspectives in population genetics ?
Line 216. Add ‘or’

Reviewer 3 ·

Basic reporting

In this study, the authors investigate the variability of different morphological traits in P. nodus among seven urban parks as a result of differences in the local enviromental pressures within a city. I think the study is interesting and the authors worked a lot. However, I think that this study is not ready to be published. The writing and the english style is correct as well as the structure of the ms, but the quality of the figures is not enough and are needed of higher resolution and the literature is too long (it has the same length that the whole paper). However, it is just a minor issue.

Experimental design

My major concern starts with the sample size and the sampling methodology. It is true that using pitfall traps allows a more random and objective sampling, but it is very important to take into consideration the sampling design. In the current explained methodology, the authors just sampled arround 160m of the total area of the park, limitating the variability of nests reaching the pitfall, and so the variability of the samples. For studing morphology, it is more reliable to carry out direct sampling instead, since it allows to collect a larger variability of workers from one same nest. And this is important (mostly in a species with different castes), since within a same nest workers differ in size. Moreover, due to task allocation within colonies, smaller workers tend to keep within the nest if their presence is not required during the foraging. So, the variability of workers collected in traps would also depend of the colony needs at that moment. Besides, 15-45 workers per park is too litle sample size. Usually, a minimum of 3-5 workers are collected per nest in monomorphic species, and 3-5 workers per caste and nest would be a reliable sample size in polymorphic species.

Validity of the findings

I fair that the variability among populations explained in the paper might be just the result of the small sample size. Although I have to say that I did not see such a difference, and groups are highly overlaping. Therefore, due to the small sample size and the explanations provided in the comments about the methodology, I can not consider that the results are reliable.

Additional comments

Unfortunately, I can not support the publication of this paper in its current status. I hope that my comments help to the authors to improve the manuscript, since I would really like to see this type of work published in the future.

Below, some detailed comments:

Introduction
The introduction is well prepared and fitting with topic. However, I have some issues with the literatura. In my opinión, the volume of the cited literatura per each statement is too large but still missing of updated and important litetarature in this topic. Here some examples: when talking about the importance of the functional traits of species, I am missing the paper published by Arnan et al. 2015, Czechowski et al. 2012, Nooten et al. 2019, Sosiak & Barden 2020, Guilherme et al. 2019…; when talking about biodiversity in urban areas I miss Brassard et al. 2022, Trigos-Peral et al. 2020...; or when talking about local adaptations due to environmental pressures I am missing the studies by Diamond (several studies), Szulking 2020. So, in general, I think that the literatura needs to be shorten and updated.

Line 68 – Please, write “Slipinski” correctly.

Table 1. I think that this table should be reconsidered. For example, head width is also a trait used as estimation of the worker size; however, my major issue is the description of the measurements – I suggest to write a more clear explanation like “máximum width of the head across the eyes”. As explanation for eye size, I would suggest that it is an estimation of the exploratory capacities of the hábitat, since larger eyes are predominant in more complex habitats and with low light. From scape length, the explanation is too narrow since it is not only used to follow trails, but also to find food, perceive vibrations, etc… I mean, this is the main sensatory organ of the ant and it is not only used for following trails. Finally, with leg length, I would also give a wider explanation since it is not only a predictor of ant speed, but also some other locomotory variables. Therefore, instead of “predictor of running speed”, I suggest tos ay something like estimator of locomotory skills (of course, keeping the part about the hábitat complexity).

Line 123-132 – it would be recomendable to include the graphical representation of the PCA, since it is quite informative. I have some concerns about the rather low variability explained by the two first PC, so actually, almost 50% of the variability is random and is not explained by the variables included in the model. One more issue is that, according to PC1 and PC2, some variables are correlated (hw-ew, sl-hlg); so it could lead to an over-representation of their values in the PCA. It can be an artifact of the statistics due to the log transformation, so I think that it would be convenient to carry out the PCA using the raw data without any transformation, or just scaling the variables. Also, why the estimation of the body size was not included? I could not find any explanation about this selection. Finally, I have some doubts in using PC3 since it only it only explains a 17% of the variability, which is almost nothing. Moreover, see my previous comments about table 1, to properly interpret the functional importance of the studied traits.

---

## Round 0.2 · Minor Revisions

Please, carefully consider the comments of the reviewer 3.

Reviewer 1 ·

Basic reporting

The authors have further improved the clarity of their study and made improvements to the literature selection based on reviewer 3s comments.

Line 108 - 116. By adding more to this paragraph, the authors created a really long sentence that became hard to read. Please split this into two or three sentences (e.g. at line 111, before "implied").

Experimental design

The authors addressed the limited sample size criticized by reviewer 2 and 3 and and improved some of their analyses and presentation of figures.
Although this was not a concern I initially raised and I am not an expert in this field, I would argue that the small sample size does remain a problem and that the authors could further improve how they present this potential confounding factor to their results in the article. As it reads now, the discussion gives the impression that the results of the study are very solid. While I think it is justified to report on the evidence derived from the statistical analyses and discuss it in the way the authors have done, I also think that the authors should more clearly add the caveat that the small sample size could be one or maybe the main reason for the differences in park populations they observed. This could be a small paragraph or at least a few sentences early in the discussion, either in the first paragraph or after the second paragraph.

Validity of the findings

no comment

Additional comments

All of my own previous concerns were addressed well by the authors and I congratulate them on their hard work.
As for the comments of the other reviewers, I think there may still be room for improvement as mentioned above, but I defer better judgement of this to the other reviewers and the editor.

Reviewer 3 ·

Basic reporting

After reading the manuscript one more time, I can not accept it for publication. Although the authors worked hard to fix the issues, in my opinion, the results of the manuscript are not fully reliable. The sample size is not enought and the sampling methodology is not correct. The authors commented that "that´s the price of using pitfall traps", which means that pitfall traps is not the correct methodology for this study. It is very important to stablish a correct methodology and a minimum sample size before starting the sampling. Still, even if they used pitfall traps (which avoid the use of different castes of the nest), the number of traps is too low for the parks site. Moreover, there is no control... so we can't accept that there is an effect of urbanization since we don't know the workers size in nature. Additionally, worker size is connected with nest size (also nest age), therefore, to test the hypothesis it is needed to start the sampling of a minimum number of nests in urban and natural areas, also the correlation between the nest size and the workers size. I know that nest collection is not always easy, so, in this case, it is needed a higher sampling effort or the selection or an easier species (I am personally familiar with the possibility of collecting 20 nest of a species in just one morning, but only 2 nests of another in one whole day). I am really sorry for my decision, but this is a 3.061 IF journal, and it requires publications that at least fit with proper sample size, appropiate methodology and reliable results.

Experimental design

As mentioned above and in my previous resutls, neither the sampling methodology or the sample size are correct. Since it has not been improved in this new version, please, read my comments in the first review.

Validity of the findings

Due to the experimental design, results are not reliable. Moreover, I asked for the graphical representation of the PCA. Having a look to it, it is visible that data from the different study sites strongly overlap and the mentioned significant different among study areas mentioned by the authors is not visible. It leads to a statistical issues.

Additional comments

I am really sorry for my decision because I see that the authors worked very hard, but I think that it is important to keep the quality of the publication according to the journal standards.

---

## Round 0.3 · accepted · Accept

The author provided proper changes after the second review.